# Influenza A, like Omicron SARS-CoV-2, Is Similarly Detected in Saliva or Nasopharyngeal Samples via RT-qPCR

**DOI:** 10.3390/v15122352

**Published:** 2023-11-30

**Authors:** Hellen Abreu, Carla Adriane Royer, Carolina Gracia Poitevin, Ana Flávia Kohler, Ana Carolina Rodrigues, Sonia Mara Raboni, Meri Bordignon Nogueira, Pedro Henrique Cardoso, Monica Barcellos Arruda, Patrícia Alvarez da Silva Baptista, Ana Claudia Bonatto, Daniela Fiori Gradia, Douglas Adamoski, Emanuel Maltempi de Souza, Jaqueline Carvalho de Oliveira

**Affiliations:** 1Department of Genetics, Federal University of Parana, Curitiba 81531-980, Brazil; ha.hellenabreu@gmail.com (H.A.); carladriane@gmail.com (C.A.R.); carol.poitevin@gmail.com (C.G.P.); ana.fk@hotmail.com (A.F.K.); ana.acr.rodrigues@gmail.com (A.C.R.); anacbonatto@ufpr.br (A.C.B.); danielagradia@ufpr.br (D.F.G.); 2Virology and Molecular Biology Research Laboratory, Federal University of Parana, Curitiba 80030-110, Brazil; sraboni@ufpr.br (S.M.R.); meribor@ufpr.br (M.B.N.); 3Institute of Technology in Immunobiology Bio-Manguinhos, Oswaldo Cruz Foundation/Fiocruz, Rio de Janeiro 21040-900, Brazil; pedro.cardoso@bio.fiocruz.br (P.H.C.); monica.arruda@bio.fiocruz.br (M.B.A.); palvarez@bio.focruz.br (P.A.d.S.B.); 4Brazilian Biosciences National Laboratory (LNBio), Brazilian Center for Research in Energy and Materials (CNPEM), Campinas 13083-970, Brazil; douglas.adamoski@gmail.com; 5Department of Biochemistry and Molecular Biology, Federal University of Paraná, Curitiba 81530-000, Brazil; souzaem@ufpr.br

**Keywords:** virus detection, diagnostic techniques and procedures, surveillance, flu, COVID-19

## Abstract

After the Coronavirus pandemic, the importance of virus surveillance was highlighted, reinforcing the constant necessity of discussing and updating the methods for collection and diagnoses, including for other respiratory viruses. Although the nasopharyngeal swab is the gold-standard sample for detecting and genotyping SARS-CoV-2 and Influenza viruses, its collection is uncomfortable and requires specialized teams, which can be costly. During the pandemic, non-invasive saliva samples proved to be a suitable alternative for SARS-CoV-2 diagnosis, but for Influenza virus the use of this sample source is not recognized yet. In addition, most SARS-CoV-2 comparisons were conducted before the Omicron variant emerged. Here, we aimed to compare Influenza A and Omicron RT-qPCR analysis of nasopharyngeal swabs and saliva self-collection in paired samples from 663 individuals. We found that both nasopharyngeal swab and saliva collection are efficient for the diagnosis of Omicron (including sub-lineages) and for Influenza A, with high sensitivity and accuracy (>90%). The kappa index is 0.938 for Influenza A and 0.905 for SARS-CoV-2. These results showed excellent agreement between the two samples reinforcing saliva samples as a reliable source for detecting Omicron and highlighting saliva as a valid sample source for Influenza detection, considering this cheaper and more comfortable alternative.

## 1. Introduction

Influenza and COVID-19 are upper respiratory tract diseases characterized by acute respiratory syndrome, which, in some cases, can progress to pneumonia, evolving into severe and lethal diseases. Both were able to advance and spread globally, generating epidemics and pandemics.

With the COVID-19 pandemic, the importance of maintaining effective virus surveillance was highlighted, reinforcing the constant necessity of discussing and updating the methods for collection and diagnoses, including for other respiratory viruses. The Influenza pandemic—2009 (Influenza A (H1N1) pdm09)—preceded COVID-19 (2019–2022, SARS-CoV-2), and the effective containment methods to fight both were the detection of infected individuals, followed by treatment, isolation to avoid spread, and mass vaccination [1,2].

Sample collection via nasopharyngeal secretion, followed by RT-qPCR, is considered the gold-standard detection method, that is, the one with the highest reliability due to factors such as sensitivity, specificity, and reproducibility; but, obtaining this sample source is uncomfortable and requires specialized teams, which can be costly and exposes healthcare workers. During the COVID-19 pandemic, many descriptions including non-invasive saliva samples were utilized as suitable alternatives for SARS-CoV-2 diagnosis [3,4,5,6,7]. On 3 May 2021, the European Centre for Disease Prevention and Control [8] published a guide including considerations for using saliva as a sample material for COVID-19 testing.

For Influenza virus diagnosis, saliva samples must still be evaluated as a feasible sample source. Additionally, detecting the more-recent Omicron variant and sub-lineages may be included in comparing paired nasopharyngeal and saliva samples to confirm the maintenance of saliva samples as a reliable sample source for diagnosis.

In this context, the study aimed to compare a multiplex RT-qPCR for detecting SARS-CoV-2 (SC2) and Influenza A (INFA), comparing paired saliva and nasopharyngeal samples (NPS) in a large group of symptomatic individuals (n = 663).

## 2. Materials and Methods

### 2.1. Sample Collection and RNA Extraction

Saliva and NPSs from 663 individuals were collected from people with mild respiratory symptoms in the LIGH-UFPR laboratory (Curitiba, Parana, Brazil—lat. −25.4470841618201, long. −49.23252521243201), between May 2022 and August 2022.

Sample collection occurred in the morning, respecting a minimum fasting time without food and drink for at least 30 min before collection. Saliva samples were self-collected using a plastic drinking straw to transfer to a prelabeled 2.0 mL microtube. For the collection of nasopharyngeal secretions, a trained team performed the collection using nasopharyngeal swabs packed in a viral transport medium (3 mL of tryptose phosphate broth from BD Bacto TM, Sparks, MD, USA). The samples were kept at 4 °C until processing in the biosafety level II laboratory.

NPSs were homogenized, whereas saliva samples underwent homogenization via vortex and centrifugation for 2 min at 2000× *g*. Aliquots of 100 µL were transferred to deep-well plates. The RNA extraction was performed using an automated magnetic EXTRACTA–RNA and DNA viral kit, according to the manufacturer’s instruction (Loccus Biotecnologia, Sao Paulo, Brazil).

### 2.2. Viruses Molecular Detection

RT-qPCR was performed to detect SARS-CoV-2 and Influenza RNAs in QuantiStudio 5™ (Thermo Fisher Scientific Inc., Waltham, MA, USA) with INFA/INFB/SC2 Bio-Manguinhos Molecular Assay (Rio de Janeiro, Brazil) used according to the manufacturer’s instructions. The test allows the detection of specific sequences in RNA extraction of target INFA—M gene (FAM), INFB—NS1 gene (VIC), SC2—N gene (CY5), and internal control—RP (human RNAse P) as endogenous gene (ROX). For negative diagnosis, only the human endogenous gene was detected in the reaction. For a positive diagnosis, the amplification of all targets with CT (cycle threshold) ≤ 40 was observed, considering the threshold of 20.000 to human endogenous—RP; 30.000 to SC2; 10.000 to INFA; and 20.000 to INFB.

According to the kit manufacturer, the limit of detection of this RT-PCR assay, based on PROBIT analyses (IBM SPSS, Statistics Subscription) and considering a positivity rate of 95% and a confidence interval (CI) of 95%, presented estimated sensitivity of 0.04 copies/µL (0.4 copies/reaction) for the INF A target, 0.08 copies/µL (0.8 copies/reaction) for the INF B target, and 0.17 copies/µL (1.7 copies/reaction) for the SC2 target, and the sample quantification of the panel was performed using a digital PCR technique.

SARS-CoV-2 positive results obtained in only one of the samples, NPS or saliva, were confirmed by a second test carried out by RT-qPCR in QuantiStudio 5 TM (Thermo Fisher Scientific Inc. Waltham, MA, USA) following protocol instruction from the Centers for Disease Control and Prevention (CDC) 2019-Novel Coronavirus (2019-nCoV) Real-Time RT-PCR Diagnostic Panel [9] targeting two regions (N1 and N2) of the nucleocapsid (N) gene.

### 2.3. SARS-CoV 2 Genotyping

Two multiplex RT-qPCR methods performed genotyping. The 4Plex SC2/VOC Bio-Manguinhos Molecular Assay amplifies a target region in the N gene and allows the identification of deletions (Del) S106, G107, and F108 in the gene ORF1a (nsp6) and the Del. H69 and V70 in the Spike gene. This protocol enables discrimination of variants and subvariants Alpha (B.1.1.7), Beta (B.1.351), Gamma (P1), Delta (B.1.617.2), Omicron (B.1.1.529—BA.1, BA.2, and BA.4/5). As an internal control (CI), the assay detects a region of the human endogenous gene, RNAse P (RP). Results were confirmed using the method described by Vogels et al. [10], which allows the detection of ORF1a Δ3675–3677, Spike Δ69–70 deletions, and CDC-N1 (N gene) to discern the VOCs Alpha, Beta, Gamma, Delta, and Omicron (BA.1, BA.2, and BA.4/5).

In addition, the identification of subvariants Omicron BA.4 and BA.5 was performed using a probe that discriminates ORF7B.L11F (CVCE3VH) (Thermo Fisher Scientific) and the GoTaq^®^ Probe 1-Step RT-qPCR System (Promega, Madison, WI, USA).

### 2.4. Data Analysis

Data were analyzed using GraphPad Prism 8^®^, to verify the descriptive statistics, differences, and correlation between diagnosis and cycle threshold (Ct) values from RT-qPCR from saliva and nasopharyngeal samples. Data were submitted to an analysis of normality, a paired T-test, and Mann–Whitney, Pearson’s, and Spearman correlations. In addition, the operational characteristics of the RT-PCR results were evaluated using sensitivity, specificity, and accuracy calculations. Positive rates and levels of agreement between the kits were assessed using Cohen’s kappa coefficients of agreement, with values ≤ 0.81–1.00 as almost perfect agreement.

## 3. Results

Figure 1A shows the distribution by the week of infections by COVID-19 and Influenza A from May to August 2022 in the academic population of Parana’s Federal University in Curitiba, Brazil. The peak is coherent with the lower air circulation in closed environments caused by the winter season temperature reduction, starting June 21 in the southern hemisphere. Among the 663 paired samples analyzed, 46 (6.9%) patients had positive detection in at least one sample for Influenza A, which was detected in 45 (6.8%) NSP samples and 42 (6.3%) saliva samples. Those values rendered a sensitivity of 91.1% and a specificity of 99.4% for the saliva assay for Influenza A detection, considering NSP the gold-standard.

Regarding SARS-CoV-2, 193 (29.1%) patients had positive detection in at least one sample; 181 (27.3%) in NPS and 178 (26.8%) in saliva samples (Figure 1B). Those values rendered a sensitivity of 92.3% and a specificity of 97.1% for the saliva assay for SARS-CoV-2 detection, considering NPS the gold-standard. Furthermore, the calculated agreement kappa index for Influenza A detection in the saliva and NPS is 0.938 (0.885–0.992). The same metric for SARS-CoV-2 detection is 0.905 [0.868–0.941]. For both disease detection, kappa values showed an excellent agreement between the two samples, showing that saliva can be used to investigate both respiratory viruses simultaneously using INFA/INFB/SC2 Bio-Manguinhos Molecular Assay and possibly other RT-qPCR multiplex kits.

Analyzing the SARS-CoV-2 variants over time in Figure 1C, it can be observed that all evaluated samples were defined as Omicron subvariants. When considering each sub-lineage, in May 2022, the BA.2 subvariant had a high prevalence, subsequently replaced by the BA.5 subvariant in the most recent months analyzed. Notably, June and July are typical winter months in southern Brazil with milder temperatures, and a significant incidence peak was observed for both COVID-19 and Influenza A infections.

## 4. Discussion

The present study includes information on 663 paired saliva and NPSs, with 46 INFA and 193 SC2 positive detection in at least one sample, with high agreement between the two sample types.

The usefulness of saliva for Influenza detection is still controversial in the literature. In 2001, Bilder et al. [11] found 100% concordance in H1N1 detection in saliva/NPS paired samples but included only 26 patients (with 14 positive cases). But, in 2017, Kim et al. [12] tested 236 paired NPS and saliva individuals for 16 respiratory viruses using multiplex RT-qPCR (Anyplex II RV16 detection kit, Seegene, Seoul, South Korea). They found twenty-three samples positive for Influenza, eight positive cases in saliva, with only four concordances in both sample types, with a kappa index of 0.22 (0.02–0.42).

With higher concordance, Sueki et al. [13] found an overall concordance of 95.8%, including 144 paired samples and 19.4% INF-positive cases. Also, Galar et al. [14], including 82 patients with 11 cases of Influenza A, described a detection rate of 97.6% in saliva, with a kappa (κ) value of 0.929 (95% confidence interval [CI], 0.832 to 1.0), indicating a high level of agreement.

Most of these studies include a limited number of positive samples. The biggest one compared 385 paired NPS and saliva samples: 120 positive INFA samples, with concordance rates of 93.5% (360/385) [15]. Our study analyzed information from many paired samples (n = 663), contributing to including saliva samples as feasible for INFA diagnosis.

For SARS-CoV-2 detection, saliva samples have been more widely studied [16,17,18,19,20,21,22,23,24,25,26,27], including some huge meta-analysis-reinforced saliva samples for SARS-CoV-2 diagnosis. For example, a systematic review and meta-analysis compared the diagnostic performance of various clinical sampling methods and including 16,762 respiratory samples describing high specificities (range 97–99%) and a negative predictive value (range 95–99%) among different clinical specimens [19]. A high overall concordance (92.5–95% CI: 89.5–94.7) was described after including 50 eligible studies reporting on 16,473 pairs of NPS/saliva samples.

The study conducted by Caixeta et al. [23] included 14,043 participants from 21 countries in their systematic review and meta-analysis. The results showed that saliva had an accuracy of 94.3%, specificity of 96.4%, and sensitivity of 89.2% compared to the reference tests. Furthermore, the sensitivity of saliva was 86.4% when compared to the combination of saliva and nasopharyngeal/oropharyngeal swabs (NPS/OPS) as the reference standard. Good sensibility was also confirmed in another meta-analysis that included 44 studies and a total of 8555 samples [28].

These reports demonstrate that saliva sample offers many advantages, such as increased patient comfort, reduced invasiveness, low risk of cross-infection, and scalability for large-scale testing. However, it should be noted that the overall agreement was slightly lower at 89.7% (95% confidence interval). Although nasopharyngeal swabs were still considered a superior testing specimen, the study highlights the benefits of saliva testing. It suggests that a potential decrease in accuracy may be acceptable, particularly in low socioeconomic regions.

Most of this meta-analysis includes positive cases since 2020. The Omicron variant and sub-lineages emerged in 2022, questioning the usefulness of RT-PCR in saliva as a feasible sample for diagnosing of this variant. But, also for these variants, recent studies have shown saliva samples as viable.

Migureres et al. [29] assess the diagnostic performance of saliva, nasopharyngeal swabs (NPSs), and anterior nasal swabs (ANSs) for detecting the Omicron variant of SARS-CoV-2. In 202 individuals, saliva exhibited the highest overall sensitivity (94.6%), followed by NPSs (90.2%) and ANs (82.6%). Notably, saliva demonstrated superior sensitivity (100%) in symptomatic patients tested within five days of symptom onset compared to ANs (83.1%) and NPSs (89.8%). The study findings emphasize the effectiveness of saliva-based RT-PCR as an early detection tool for the Omicron variant, offering advantages such as improved patient acceptance and the potential for limiting viral spread through earlier detection. Apparently, the Omicron variant may be detected in saliva as well as previous variants; some studies suggest this sample is superior, for example, when compared to Delta variant detection [30].

Additionally, other groups confirmed the feasibility of saliva specimens for Omicron detection, such as Uršič et al. [31], including 624 participants during the Delta and Omicron waves; Ahti et al. [32], involving 250 participants with the Omicron BA.2 variant and 135 positive cases in NPS/134 positive in saliva (kappa coefficient = 0.911; *p* = 0.763); and Bordi et al. [33], including 255 samples and 85 Omicron-positive patients.

Our group published a previous study, conducted from August 2020 to November 2020 when the original SARS-CoV-2 was circulating, comparing the detection of NSP and saliva samples [4]. The study included 229 participants, and the results showed that saliva had an 87.80% sensitivity, 98.94% specificity, and 96.94% accuracy in detecting the virus. These numbers were similar of those obtained in the present study with Omicron subvariants (92% sensitivity, 100% specificity, and 97% accuracy). As the collections were conducted in the same conditions and methods, these numbers also reinforce the feasibility of SARS-CoV-2 saliva detection for Omicron variants.

There were 13 cases of COVID-19 detected only in NPS and 11 only in saliva. All discordant samples had positive RT-qPCR confirmed by using a second methodology [9]. Three patients had COVID-19 and Influenza. Of these, two were in the nasopharyngeal swab and another only in saliva. Virus detection discordant in samples from different sources may be correlated with the progression of the disease and the fact that the presence of viral loads in the nasopharyngeal and saliva may be distinct [34].

Regarding the cycle threshold, lower Ct values were observed in NPS samples, which denoted a higher viral load for Influenza A (Figure 2A, left) and SARS-CoV-2 (Figure 2A, right). This scenario was already observed for SARS-CoV-2 testing using saliva samples [4], where higher viral loads were consistently obtained when using nasopharyngeal swabs. Furthermore, Ct values seem poorly correlated (Pearson r^2^ = 0.3252 for Influenza A and r^2^ = 0.4743 for SARS-CoV-2) between both sample types (Figure 2B).

Research groups from various fields have investigated viral pathogens in the upper airways through saliva. In 2011, Bilder et al. [11] studied 26 patients and detected the virus in both saliva and nasopharyngeal swab samples. They observed an average cycle threshold (Ct) of 22.4 for swab samples and 26.35 for saliva samples, consistent with our findings. In our study, the mean Ct for nasopharyngeal swab samples positive for Influenza A was 23.35, while for saliva samples it was 26.95.

The statistical difference for the Ct of virus target genes and RP was found (low CTs and higher RNA detection in all targets). However, it is relevant to report that there were 10 cases in which the Ct of the Influenza target gene in the NPS positive-to-Influenza sample was higher than the Ct of the saliva sample, indicating a lower viral load in the former. For the SARS-CoV-2 target gene, there were 28 cases where the Ct of the saliva sample was lower than the Ct of the NPS positive sample. This also demonstrates the sensitivity and effectiveness of saliva self-testing, particularly in the context of the need to analyze a large number of individuals with mild to moderate symptoms to track positive cases and isolate them in an effort to reduce the spread of the disease.

Considering reasons for differences between the sensibility of saliva and NPS virus detection, it is important to consider the Ct values differences that were found low in NPS samples; on the other hand, this is not the unique reason for the differences, considering the number of cases where detection in saliva samples had a low Ct value when compared to NPS and the low correlation between Cts in saliva and NPS. So, it is also important to consider that the virus detection discordant in samples from different sources may be correlated with the presence of viral loads in the nasopharyngeal and saliva samples in various stages of disease and different individuals [34,35].

Working with RNA detection, it is crucial to consider the time interval between sample collections, processing, RNA extraction, and RT-qPCR analysis, as this can influence the identification and quantification of the viral load present [11]. It is known that time and storage conditions can interfere with RNA abundance. For example, even 20 min at room temperature may the affect integrity of endogenous salivary β-actin mRNA [36]. Specifically, for SARS-CoV-2 detection in saliva, viral RNA was demonstrated to be stable at 4 °C, room temperature (~19 °C), and 30 °C for prolonged periods (more than 15 days) [37]. SARS-CoV-2 RNA stability at room temperature without the need for expensive cooling strategies is quite interesting, which is especially beneficial in regions or countries with limited resources and collection away from laboratory structure.

Although the possibility of stability of viral RNA in saliva is known, it is important to emphasize that, in the present study, the saliva was processed under optimal conditions (less than 3 h) and, in this condition, the sensitivity was high. It is crucial to consider the time interval between sample collections, processing, RNA extraction, and RT-qPCR analysis, which may influence the identification and quantification of the viral load present, affecting sensibility values. But, this variant was not tested in the present study.

From a scientific perspective, SARS-CoV-2 and Influenza in both saliva and NPSs are highly relevant as they provide insights into the mechanism of viral replication within the human body. Coronaviruses, including SARS-CoV, are primarily associated with respiratory tract infections. This is due to the Spike protein of the virus, which binds to angiotensin-converting enzyme-2 receptors (ACE-2) from the cell membrane. ACE-2 receptors are predominantly expressed in respiratory tract cells. However, they can be expressed in other tissues such as the lungs, kidneys, small intestine, testes, thyroid, and adipose tissue [10]. The SARS-CoV-2 underwent evolutionary changes and continues to evolve. It acquired affinity and adhesion to another cellular receptor, the cell surface serine protease receptor TMPRSS2, expressed in the respiratory tract cells. This interaction between the virus and cellular receptors contributes to its presence and replication within the respiratory tract region. Understanding these mechanisms is crucial for comprehending the pathogenesis and transmission of coronaviruses [38].

Influenza A viruses are equipped with a functional binding protein, HA (hemagglutinin), which also uses TMPRSS2 and HAT (trypsin-like protease) to attach to the host cell. Human INFA has preference for SA (sialic acid on sialylated glycoconjugates) linked to the glucose residue through an α-2,6 bond. The abundance of these receptors in humans is highest in the nasopharynx, bronchi, and lungs [39,40,41]. Although with different mechanisms for virus cell entrance, the present study describes the relevance of considering saliva samples for diagnosis of both virus types.

It is important to explore this type of data and contribute to alternatives for virus diagnosis, as the recent experience with the pandemic has highlighted the fragility and need for knowledge of approaches that meet scenarios such as this one, where there is a global shortage of medical supplies for high and unexpected demand, as well as hand-skilled and trained workforce. The nasopharyngeal secretion collection test is highly sensitive, but the risks arising from this invasive technique are not discarded, which, if poorly conducted, can lead to a false negative result [42]. In this context, self-collection of saliva shows adequate performance.

Confirmational comparisons in multiple publications are essential for validating methodologies in large sample sizes and in differential realities. This requires significant investment and time due to the large volume of data generated. Nevertheless, the studies focused on local realities are relevant due to population particularities and the validation of methods in clinical practices, leading to the generation of final official reports.

## 5. Conclusions

Our results demonstrated the efficient detection of Influenza A and Omicron BA.2, BA.4, and BA.5 SARS-CoV-2 via RT-qPCR in saliva and nasopharynx samples, with high sensitivity and accuracy (>90%) and with kappa indexes of 0.938 (0.885–0.992) and 0.905 (0.868–0.941), respectively. These results reinforce the good performance of saliva compared with gold-standard NPS swabs.

The feasibility, efficacy, sensitivity, and accuracy of diagnosing COVID-19 and Influenza A via INFA/INFB/SC2 Bio-Manguinhos Molecular assay (Rio de Janeiro, Brazil) were proved by experiments conducted in this study and also highlight saliva as a valid sample source for Influenza detection, considering this cheaper and more comfortable alternative.

## Figures and Tables

**Figure 1 viruses-15-02352-f001:**
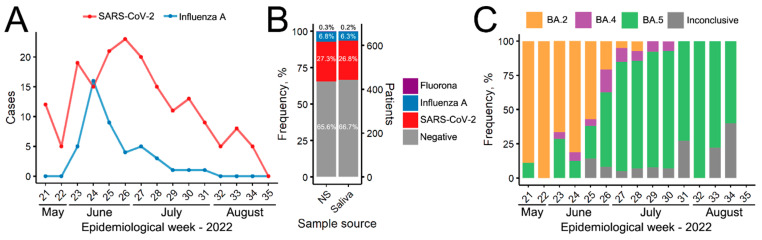
Multiplex detection of SARS-CoV-2 and Influenza in nasopharyngeal swabs and saliva samples: (**A**) Positive cases of Influenza A and COVID-19 detected by epidemiological week from 2022. Values combine patients’ positive in NSP, saliva, or both. (**B**) Frequency of patients positive for Influenza A, SARS-CoV-2, Fluorona (coinfection), or negative diagnosis in each type of sample source. (**C**) Frequency of Omicron subvariants by epidemiological week.

**Figure 2 viruses-15-02352-f002:**
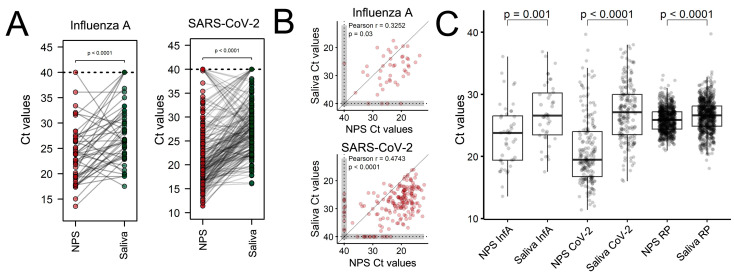
Comparison of cycle threshold values between saliva and NS samples: (**A**) Comparison of CT from RT-qPCR of NPS and saliva to Influenza A (left) and SARS-CoV-2 (right). *p*-values derived from the Welch t-test, each dot is a patient, and lines connect samples from the same patients. Cases with no amplification were defined in Ct 40. (**B**) Correlation between Ct value from NPS and saliva for Influenza A (above) and SARS-CoV-2 (below). Correlations values derived from Pearson correlation; gray areas denote no amplification in one of the samples; and the line represents the theoretical curve for perfect correlation. (**C**) Comparison of CT from RT-qPCR of NPS and saliva for Influenza A, SARS-CoV-2, and human RNase P targets. *p*-values derived from the Welch *t*-test, the dark line is the median, the box extends from the first to the third quartile, and each dot is a single diagnosis.

## Data Availability

Data are contained within the article.

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
