# Peer review of "Influenza A, like Omicron SARS-CoV-2, Is Similarly Detected in Saliva or Nasopharyngeal Samples via RT-qPCR"

_viruses, 2023, doi:10.3390/v15122352_

Round 1

Reviewer 1 Report

Comments and Suggestions for Authors

After the COVID-19 pandemic, the need for improved virus surveillance methods became evident. While nasopharyngeal swabs are standard for detecting SARS-CoV-2 and influenza, they are often uncomfortable for patients. During the pandemic, saliva samples were shown as an effective alternative for SARS-CoV-2 diagnosis, though for influenza they were not still tested widely. This study aimed to compare a multiplex RT-qPCR for detecting SARS-CoV-2 (SC2) and Influenza A (INFA) using paired saliva and nasopharyngeal samples (NPS) from 663 symptomatic individuals. Both methods had over 90% accuracy for detecting Omicron and Influenza A virus. The results confirm saliva might be considered as a cost-effective and comfortable sample source for both viruses. 

I need to say that this is really a fairly simple but useful study to evaluate the possibility of using saliva for testing, instead of nasopharyngeal swabs.

Some minor questions:

- What still would be extremely interesting is to elaborate on the reasons for the small differences between the two approaches. Are the differences caused by high values of the threshold cycle or something else?

- Another question is how did the rapid processing time in this study (less than 3 hours) influence the results? What if the samples will be transported for a longer period of time.

Author Response

According to reviewer 1:

After the COVID-19 pandemic, the need for improved virus surveillance methods became evident. While nasopharyngeal swabs are standard for detecting SARS-CoV-2 and influenza, they are often uncomfortable for patients. During the pandemic, saliva samples were shown as an effective alternative for SARS-CoV-2 diagnosis, though for influenza they were not still tested widely. This study aimed to compare a multiplex RT-qPCR for detecting SARS-CoV-2 (SC2) and Influenza A (INFA) using paired saliva and nasopharyngeal samples (NPS) from 663 symptomatic individuals. Both methods had over 90% accuracy for detecting Omicron and Influenza A virus. The results confirm saliva might be considered as a cost-effective and comfortable sample source for both viruses.

I need to say that this is really a fairly simple but useful study to evaluate the possibility of using saliva for testing, instead of nasopharyngeal swabs.

 Some minor questions:

- What still would be extremely interesting is to elaborate on the reasons for the small differences between the two approaches. Are the differences caused by high values of the threshold cycle or something else?”

Response: We are grateful for the relevant comment. The study is simple but greatly useful, showing the possibility of including saliva as a feasible sample also for Influenza diagnosis.

                Considering the reasons for the slight differences in sensibility between the two sample types, it is important to consider that, in general, the cycle threshold (Ct) was lower for the SARS-CoV gene target and for the Influenza gene target in nasopharyngeal samples. Initially, this suggests that the small differences in sensibility may be explained, at least in part, by the Ct values differences.

                On the other hands, this is not the unique reason for the differences.  For example, for influenza detection, there is a positive sample in saliva that is not positive in NS, and this sample has an impressive low Ct value (Ct=27) (Figure 2A). For SARS-CoV-2, there are many other examples, considering 8 cases that were positive only in saliva. There are some examples of high Ct values (>35), but there are also Cts of 25. This is true also considering positive samples only in NS.

                Considering these examples with low Ct values and detection in only one sample type, it is important to consider that the virus detection discordant in samples from different sources may be correlated with the progression of the disease and the fact that the presence of viral loads in the nasopharyngeal and saliva may be distinct [37].

                We include this discussion in the manuscript.

“- Another question is how did the rapid processing time in this study (less than 3 hours) influence the results? What if the samples will be transported for a longer period of time.”

Response: In the present study, the samples were promptly transported to the laboratory, processed, extracted, and subjected to RT-qPCR within less than 3 hours.

It is known that time and storage conditions can interfere with RNA abundance. For example, even 20 minutes at room temperature may the affect integrity of endogenous salivary β-actin mRNA (Park et al., 2006). Specifically, for SARS-CoV-2 detection in saliva, viral RNA was demonstrated to be stable at 4°C, room temperature (~19°C), and 30°C for prolonged periods (more than 15 days) (Ott et al., 2021). SARS-CoV-2 RNA stability at room temperature without the need for expensive cooling strategies is quite interesting, which is especially beneficial in regions or countries with limited resources and collection away from laboratory structure.

Although the possibility of stability of viral RNA in saliva is known, it is important to emphasize that, in the present study, the saliva was processed under optimal conditions and, in this condition, the sensitivity was extremely high. It is crucial to consider the time interval between sample collections, processing, RNA extraction, and RT-qPCR analysis, which may influence the identification and quantification of the viral load present, affecting sensibility values. But this variant was not tested in the present study.

We also include this discussion in the manuscript.

References:

Park, N. J.; Li, Y.; Yu, B.M.N.; Brinkman, T.; Wong, D.T. Characterization of RNA in Saliva. Clinical Chemistry. 2006, 52 (6), 988–994. https://doi.org/10.1373/clinchem.2005.063206.

Ott, I.M.; Strine, M.S.; Watkins, A.E.; Boot, M.; Kalinich, C.C.; Harden, C.A.; Vogels, C.B. F.; Casanovas-Massana, A.; Moore, A.J.; Muenker, M.C., et al. Stability of SARS-CoV-2 RNA in Nonsupplemented Saliva. Emerg Infect Dis. 2021, 27(4), 1146-1150. doi: 10.3201/eid2704.204199.

Reviewer 2 Report

Comments and Suggestions for Authors

The authors present the results obtained on a study of more than 600 samples aimed at simultaneously identifying the presence of the INFA virus and the Omicron variant of SARS-CoV-2. 

The authors emphasise the efficiency of the test from saliva samples, which unlike swabs or other biological material is less invasive. Several works in the literature use the same biological material for the identification of the above-mentioned viruses. Unlike these, the novelty is in the simultaneous identification. Thus, the work has the advantage of being able to diagnose two infections sustained by different viruses. The limitation is the high mutational variability that characterises RNA viruses, as we have seen following the emergence of the different variants of SARS-CoV-2. This could affect the outcome of the RT-PCR assay. Nevertheless the work could be published following revisions. 

-Lines 45-47: please try to change the sentence; there is redundancy in the term "surveillance". 

-With regard to the RT-PCR assay, have tests been set up to assess the Limit of detection (LoD)? If not, would it be possible to test the identification capability of this assay? 

- I suggest reducing the discussion section, especially in the parts where reference is made to results obtained in other works.

Author Response

According to reviewer 2:

“The authors present the results obtained on a study of more than 600 samples aimed at simultaneously identifying the presence of the INFA virus and the Omicron variant of SARS-CoV-2.

The authors emphasise the efficiency of the test from saliva samples, which unlike swabs or other biological material is less invasive. Several works in the literature use the same biological material for the identification of the above-mentioned viruses. Unlike these, the novelty is in the simultaneous identification. Thus, the work has the advantage of being able to diagnose two infections sustained by different viruses. The limitation is the high mutational variability that characterises RNA viruses, as we have seen following the emergence of the different variants of SARS-CoV-2. This could affect the outcome of the RT-PCR assay. Nevertheless the work could be published following revisions.

-Lines 45-47: please try to change the sentence; there is redundancy in the term "surveillance".

Response: We are grateful for the relevant comment and we changed the sentence (highlighted in main manuscript).

-With regard to the RT-PCR assay, have tests been set up to assess the Limit of detection (LoD)? If not, would it be possible to test the identification capability of this assay?

Response: According to the kit manufacturer, the Limit of detection of this RT-PCR assay, based on PROBIT analyses (IBM SPSS Statistics Subscription) and considering a positivity rate of 95% and a confidence interval (CI) of 95%, presented estimated sensitivity of 0.04 copies/µL (0.4 copies/reaction) for the INF A target, 0.08 copies/µL (0.8 copies/reaction) for the INF B target, and 0.17 copies/µL (1.7 copies/reaction) for the SC2 target, and the sample quantification of the panel was performed using digital PCR technique. Information available at https://www.bio.fiocruz.br/images/bm-bul-158-01-r-sn---mol-infa-infb-sc2--.pdf.

                We included this information in the manuscript (“2.2 Viruses molecular detection” item).

- I suggest reducing the discussion section, especially in the parts where reference is made to results obtained in other works.”

Response: We are grateful for the relevant comment, and we improved the discussion section. Although suggestion, we maintained most of the results obtained in other works considering that this information is important to support our findings.